# Real-time racial discrimination, affective states, salivary cortisol and alpha-amylase in Black adults

**Soohyun Nam**[1]*, **Sangchoon Jeon**[1], **Soo-Jeong Lee**[2], **Garrett Ash**[3,4], **LaRon E. Nelson**[1], **Douglas A. Granger**[5]

**1** Yale University, School of Nursing, Orange, CT, United States of America, **2** School of Nursing, University of California, San Francisco, San Francisco, CA, United States of America, **3** Veterans Affairs Connecticut Healthcare System, West Haven, CT, United States of America, **4** Center for Medical Informatics, Yale School of Medicine, New Haven, CT, United States of America, **5** School of Social Ecology, Institute for Interdisciplinary Salivary Bioscience Research, University of California, Irvine, Irvine, CA, United States of America

* soohyun.nam@yale.edu

## Abstract

Perceived racial discrimination has been associated with the autonomic nervous system (ANS) and hypothalamic-pituitary-adrenal (HPA) axis activities—two major stress response systems. To date, most studies have used cross-sectional data that captured retrospective measures of the racial discrimination associated with current physiological stress responses. The purpose of this study was to examine the relationship between racial discrimination measured in real-time and physiological stress responses. Twelve healthy Black adults completed baseline surveys and self-collected saliva samples 4x/day for 4 days to measure cortisol and alpha amylase (AA) as a proxy of HPA and ANS systems, respectively. Real-time racial discrimination was measured using ecological momentary assessments (EMA) sent to participants 5x/day for 7 days. Multilevel models were conducted to examine the relationship between racial discrimination and stress responses. In multilevel models, the previous day's racial discrimination was significantly associated with the next day's cortisol level at wakening ($\beta = 0.81$, partial $r = 0.74$, $p<0.01$) and diurnal slope ($\beta = -0.85$, partial $r = -0.73$, $p<0.01$). Also, microaggressions were significantly associated with the diurnal cortisol slope in the same day, indicating that on the day when people reported more microaggressions than usual, a flatter diurnal slope of cortisol was observed ($\beta = -0.50$, partial $r = -0.64$, $p<0.01$). The concurrent use of salivary biomarkers and EMA was feasible methods to examine the temporal relationship between racial discrimination and physiological stress responses. The within-person approach may help us understand the concurrent or lagged effects of racial discrimination on the stress responses. Further studies are needed to confirm the observed findings with a large sample size and to improve stress related health outcomes in racial/ethnic minorities.

**Data Availability Statement:** All relevant data are within the paper and its Supporting Information files.

**Funding:** The current study was funded by Yale school of nursing research program. The grant number was not assigned. The sponsor did not play any role in the study design, data collection and analysis, decision to publish or preparation of the manuscript.

**Competing interests:** Douglas Granger is founder and chief scientific and strategy advisor at Salimetrics LLC and Salivabio LLC and these relationships are managed by the policies of the committee's on conflict of interest at the Johns Hopkins University School of Medicine and the University of California at Irvine. The other authors have no conflicts of interests. This does not alter our adherence to PLOS ONE policies on sharing data and materials.

## Introduction

Structural racism operates through a mutually reinforcing system that includes historical, cultural, institutional and interpersonal racism that inequitably limits opportunities for social and economic advancement for people of color and perpetuates racial group inequity [1–3]. Thus, psychological stressors that racially minoritized groups experience are multifactorial, often compounded by racial issues and other socioeconomic strains [4]. Social stress deriving from systems of inequality such as racial discrimination may provoke a severe psychological and physiological response and has been shown to have a profound effect on the functioning of the autonomic nervous system (ANS) and hypothalamic-pituitary-adrenal (HPA) axis [4, 5]. Research has shown that the ANS and HPA systems are potential mediators between psychological stress and poor health and might play a critical role in disparate morbidity and mortality among racially minoritized adults such as Black Americans (hereinafter, Black adults) [6, 7].

ANS activation is described as the "defense reaction" (a.k.a "fight-or-flight response") that enhances cardiovascular tone, respiratory rate, and elevates blood glucose [8]. ANS activation reflects the characteristics of the situation (e.g., controllable stressors) and the individual (e.g., Type A personality) [8]. In contrast to the ANS, HPA activation is described as the "defeat reaction," a passive response pattern characterized by emotional distress, behavioral withdraws or avoidance, and loss of control [8]. Studies have supported that HPA axis activation is likely to occur in stressful situations that are uncontrollable [9]; thus, those with high HPA reactivity are more likely than those with low reactivity to perceive themselves as having little control in life [10]. Also, the effect of stress on the HPA system depends on the type, chronicity and severity of the stressors [11]; for example, chronic stress is associated with lower morning and higher evening cortisol levels, resulting in a flatter diurnal slope [11–13], which has been associated with chronic disease [14, 15].

To date, the HPA axis is the most widely studied neuroendocrine stress system in the context of stress research, and cortisol has been studied as a gold standard biomarker to evaluate the HPA axis [16]. Alpha-amylase is a salivary enzyme that helps to break down carbohydrates. Salivary alpha-amylase (sAA) is secreted by the parotid glands in response to adrenergic activity by stress-triggered locus coeruleus firing, and has been used as a valid and reliable surrogate marker of ANS activities in stress research [17]. The diurnal secretion pattern of sAA is opposite to the diurnal rhythm of cortisol. That is, while the diurnal cortisol pattern is high in the morning and decreases throughout the day, the sAA levels decrease in the first 30 minute after awakening and increase throughout the day [18]. A growing number of studies have shown that potentially asymmetric and reciprocal relations between the HPA and ANS systems may have implications related to the emotional and behavioral responses to stress and pathogenesis of stress-related disease [7, 19, 20]; however, little is known about diurnal sAA profiles in response to racial discriminations.

Racial discrimination has been predictive of an increased negative affect among Blacks, which links to unhealthy lifestyle behaviors [21, 22]. ANS activity, in particular, has been shown as a major component of the emotional responses—positive and negative affect—in certain clinical states and healthy populations [23]. Given that the ANS and HPA systems may be differently activated in response to different situational demands as well as by individuals' perceptions of stressful events [24], examining ANS and HPA axis activities and affective states may shed light on our understanding of the effect of racial discrimination on biological stress reactivity and on accompanying behavioral responses.

To date, findings of studies on the effect of racial discrimination on the stress systems are inconclusive. One possible reason is that most research in racial discrimination has focused

primarily on between-person differences (vs. within-person process) in stress exposure and HPA axis reactivity. Data from a growing number of studies, however, suggest that over 50% of the variability in cortisol is attributable to within-person, within-day variances [25, 26]. The ups-and-downs of within persons daily racial discrimination experience may be associated with ANS or HPA system activities on the same day or on the next day among Blacks adults; however, it has yet to be examined. For example, studies with other populations have shown that stressful events influence individuals' cortisol levels the next morning [27, 28]. Along similar lines, children secreted more cortisol at wakening following days when they experienced more peer rejection incidents than their usual [29, 30].

Another potential factor that may have contributed to the inconclusive findings across the studies is the method of examining temporal relationships between racial discrimination and ANS and HPA system activities. Many studies on racial discrimination in Black adults have used a cross-sectional study design or data derived from a single wave of a longitudinal study that captured the retrospective measures of the racial discrimination associated with individuals' current HPA system without specifying the time frame in which the individual experienced racism-related psychological stress. Such retrospective measures may be subject to recall and rumination bias. In addition, the nomothetic approach may reflect stable between-person characteristics in racial discrimination rather than a temporal relationship between triggers and the stress responses [31].

A prospective study design with real-time data can help examine the effect of racial discrimination on biological stress responses and within-person fluctuations. This approach appears particularly helpful for examining a subtle form of racial discrimination—'microaggressions.' For example, when individuals are uncertain or traumatic about the meaning of experiences from ambiguous or overt discrimination, they may be more likely to have a delayed response (i.e., lagged effects) from rumination [31]. Within-person data allow for examining concurrent and lagged effects of racial discrimination on stress systems and for eliminating potential confounders between individuals (e.g., sex, comorbidities). However, no published studies have examined diurnal cortisol and sAA profiles together, incorporating real-time, racial discrimination in a natural living environment in Black adults.

The purpose of the study was to: (a) examine the within-person effects of the same day- and previous-day racial discrimination/microaggression on HPA activity (cortisol indexes: Area Under the Curve [AUC], awakening response, at wakening, diurnal slope), (b) examine the within-person effects of the same day and previous-day racial discrimination/microaggression on ANS function (sAA indexes), and (c) examine the within-person effects of the negative affect on ANS and HPA system activities. Due to the inconsistent research findings on the effect of racial discrimination on cortisol and lack of data on sAA, we did not make a priori hypotheses regarding the direction of the effect (increased vs decreased) in the morning vs. evening. Instead, we hypothesized that Black adults experiencing more racial discrimination or negative affect than usual would experience increased total outputs of sAA or cortisol (AUC) on the same day or next day (lagged effect) in the within-person analyses. With the exploratory nondirectional hypotheses, we examined the within-person awakening response (e.g., decline in sAA and incline in cortisol after awakening), and diurnal slope (e.g., flatter cortisol slope means lower morning cortisol and higher bedtime cortisol).

## Methods

### Study design

This study was a prospective, observational design to examine the effects of real-time racial discrimination on biological stress responses in Black adults. Each participant served as his/her

own control to assess the within-person effects of racial discrimination on repeatedly measured salivary stress biomarkers [32]. Within-person analysis of the effects of racial discrimination was at the day level using average scores of Ecological Momentary Assessment (EMA) response across the day. EMA is a real-time data capture strategy that allows researchers to collects self-report data in real-world settings where people engage in their typical routines in their everyday lives.

## Participants and recruitment

Participants were recruited from northeastern urban communities. We collaborated with African American church communities who partner with university researchers for community-engaged research. Details of the community-engaged process and overall study protocols were published elsewhere [33]. Inclusion criteria were as follows: (a) self-reported African American or Black, (b) 30–55 years old, (c) ownership of a smartphone, (d) currently employed, (e) able to respond to at least 75% of the EMA prompts sent, and (f) English-speaking. We excluded those who were: (a) pregnant, (b) had serious acute or terminal medical conditions or other conditions that confound salivary biomarker outcomes (e.g., radiation of salivary glands, Cushing's or Addison's disease) [34], (c) were using current corticosteroid or cytokine-based treatment [34], or (d) shift-workers who may have variations in circadian rhythms that affect diurnal rhythms in ANS and HPA system [35]. The study focused on young- and middle-aged, working Black adults who were more likely to be exposed to workplace racial discrimination.

For power analysis, we determined the effect sizes based on cross-sectional correlations of predictors and intraclass correlations (ICC) of outcomes measured on repeated occasions within the same individual. Based on a previous study [36], we calculated an ICC of 0.42 on cortisol AUC with a design effect (DE) of 3.52 (DE = 1+0.42*[4–1]) on the various cortisol indexes measured for 7 days. By adjusting for the DE of 3.52, the expected 84 (= 12 people X 7 days) observations from 12 participants will have equivalent power with 24 independent samples. Thus, the sample size of 12 can detect the medium effect sizes of 0.53–0.60 with approximately 80%-90% power at a 5% significance level.

## Procedures

We obtained Institutional Review Board approval from Yale University and written informed consent from all participants. After conducting screening interviews by phone, we collected information on participants' sleep, wake, and commuting schedules to accommodate EMA delivery time. The mEMA® app (ilumivu.com) is compatible with both iOS and Android operating systems. At the baseline visit, we loaded the mEMA app into each participant's smartphone. During the initial visit, we provided one-on-one, in-person training to all participants regarding how to respond to EMA surveys. Then, participants were asked to respond to the EMA survey prompted at a random time within five pre-programmed windows daily (signal-contingent sampling; e.g., 7–9 am, 11am-12pm, 1-3pm, 4–6 pm, 7–9 pm) for seven days (a total of 35 signals). We administered baseline surveys with participants and measured each participant's body weight and height using a portable electronic scale (Omron HBF-514C®) and a stadiometer (Seca®) following standard procedures. Body mass index (BMI) was calculated as weight (kg)/height squared ($m^2$). After five minutes rest, blood pressure was measured twice with an automated cuff (Omron HEM 780 IntelliSense®), with one minute between readings; the average of the two readings was recorded.

**Salivary biomarkers collection and determinations.** After we provided one-on-one training and demonstration for saliva sample self-collection and detailed instructions on

storage, participants received 16 labeled, color-coded cryovial collection devices. Participants were asked to fill unstimulated whole saliva via passive drool to a designated line (1.5ml) using a collection aid, for a total of 60–90 sec four times per day (at wakeup, 30 min after wakeup, before dinner, and at bedtime) (SalivaBio, Carlsbad, CA) for four consecutive days, and to store the samples in their home freezer until research staff picked up the samples [37]. We instructed participants not to eat or to only drink water within 60 minutes prior to saliva collection, and not to brush their teeth, smoke, or exercise for 30 minutes prior to saliva collections.

We sent reminders prompted by EMA, 4x/day to participants to collect each saliva sample, record saliva collection dates and times in collection kits, and send us a picture of the collected kit with the time and date on it through EMA. After the 7-day collection period, research staff transported the frozen saliva samples from the participants' homes in a cooler with ice packs. The collected samples were frozen at -80˚C in the freezer at our bio-behavioral lab. The frozen samples were shipped overnight to the Institute for Interdisciplinary Salivary Bioscience Research, where the samples were assayed in a single batch. On the day of assay, the samples were thawed, centrifuged (to remove mucins) and assayed in duplicate for cortisol and sAA using commercially available assays specifically designed for use with saliva (Salimetrics, Carlsbad, CA) per the manufacturer's recommended protocols. For all assays, the inter- and intra-assay coefficients of variation were, on average, less than 15% and 10%, respectively. Collection of saliva multiple times a day permitted assessment of the within-day effect of key components of diurnal cortisol and sAA rhythms (i.e., a total output, awakening response, the diurnal slope).

**Baseline measures.** Baseline surveys included sociodemographics, current smoking status (yes/no), and alcohol consumption by the Alcohol Use Disorders Identification Test [38], which included the frequency of drinking and amount of alcohol consumption. We also used the following validated self-report measures collected at baseline:

Racial discrimination was measured using the *Major Life Discrimination* (MLD) [39] and the *Race-Related Events Scale* (RES) scales [40]. The MLD scale is a 9-item self-report measure of past exposures to lifetime discrimination. Respondents indicated whether they had ever experienced each listed major discrimination event (e.g., denied a bank loan, unfairly fired, getting declined for a job, at work, stopped by police) (Cronbach's α = >0.88) [39]. The sum MLD score ranged from 0 to 9 from each yes/no item; higher scores indicated more lifetime discriminatory experiences. The RES had 22 items to measure exposure to stressful and potentially traumatizing race-related experiences by responding "yes/no." The total RES score ranged from 0 to 22 (Cronbach's α = 0.78–0.88) [40]; higher scores indicated more experiences of race-related stressful events.

*The Black Racial Identity–Centrality subscale* (α >0.77) is an 8-item, 7-point Likert scale (strongly disagree = 1 to strongly agree = 7) that measures the extent to which individuals normatively define themselves with regard to race. It is a measure of whether race is a core part of an individual's self-concept [41]. After reverse-scoring three items, an overall score was calculated by averaging all items, with higher scores indicating stronger Black racial identity.

*Subjective Social Status*. Participants were asked to place an '*X*' on the rung that best represented where they thought they stood on the ladder, with the 10 rungs described as follows: At the bottom are people who are the worst off, those who have the least money, least education, and worst jobs or no job. At the top of the ladder are people who are the best off, those who have the most money, most education, and best jobs. (Test-retest reliability, $\rho$ = 0.62) [42].

*The Center for Epidemiological Studies Depression Scale (CES-D)* is a 20-item, 4-point Likert scale that captures current depressive symptoms: how respondents have felt or behaved during the last week by selecting one of four options (0 = Rarely, 1 = Some of the time,

2 = Occasionally, 3 = Most of the time). The score ranges from 0 to 60 and higher scores indicate greater depressive symptoms (Cronbach's α >0.85) [43].

## Ecological Momentary Assessment measures

Daily racial discrimination was measured by the Experiences of Discrimination (EOD) (α>0.88) [39] and Racial Microaggressions Scale (RMAS) (α>0.85) [44, 45] adapted for EMA surveys. Because the EOD and RMAS measure experiences of unfair treatment over the past month to year, of which response options are not relevant for the real-time EMA assessment, we revised the question wording and response choices for the EMA time frame using yes/no answers or Likert scale options. Subscales of the EOD included worry, global, filed complaint, response to unfair treatment, day-to-day discrimination, and skin color [39]. Subscales of the RMAS included invisibility, criminality, low-achieving/undesirable culture, sexualization, foreigner/not belonging, and environmental invalidations [44]. Participants were asked to report whether they had experienced any unfair treatment from a list of 11 common daily racial discriminations since their last prompt or within the last 2–3 hours if they missed or did not complete their last prompt (e.g., "treated with less courtesy than other people because of your race or ethnicity," yes = 1/no = 0) and also from a list of 32 microaggression experiences (e.g., "people mistake me for being a service worker simply because of my race or ethnicity," 1 = strongly disagree to 7 = strongly agree). Possible daily scores of the EOD ranged from 0 to 10, with higher scores indicating more racial discriminatory experiences. Possible daily scores of the RMAS ranged from 15 to 105, with higher scores indicating greater experiences of microaggressions.

Affective States were measured by the extent to which participants felt eight different types of emotions at the moment of the prompt. A ten-point response scale ranging from "not at all" to "extreme" (e.g., "How tense or anxious do you feel right now?") was used for each item. A negative affect variable was created by averaging the scores for each of the following items: emotionally upset, stressed, annoyed/angry, lonely/alone, sad/depressed, tense/anxious, and discouraged/frustrated (Cronbach's α = 0.85) [46]. The positive affect measure consisted of one item (i.e., happy).

**Salivary biomarker measures.** The four collection times permitted assessment of within-day effects of key components of the diurnal sAA and cortisol rhythms. Total daily cortisol and sAA outputs were measured using the AUC. AUC with respect to ground (AUCg) was calculated using the trapezoid formula [47]. Awakening responses were calculated by subtracting the 30 minutes after-wake sample from the wake sample, and multiplying by half of the time between samples. To calculate the diurnal slope, the bedtime sample was subtracted from the wake sample, and this was divided by the time between samples.

**Data analysis.** Descriptive analysis was performed for demographic characteristics and baseline assessments with mean, standard deviation (SD), and frequency. We computed the Spearman's correlation coefficients between the baseline characteristics and the average of daily measured EMA/salivary biomarkers. A Wilcoxon's test was performed to examine baseline characteristics and the average of daily measured EMA/salivary biomarkers by sex. To examine within-person effects on daily salivary stress biomarkers, multilevel models were performed to predict each of the cortisol and sAA metrics (AUC, awakening response, at wakening, and diurnal slope) with three EMA measures including racial discrimination, microaggressions, and negative affect. All continuous variables were standardized for zero mean and one standard deviation. Multilevel models included a random intercept for each subject and incorporated within-subject correlations. The within-person effects were examined for the EMA survey on the previous day (lagged effect) and on the same day of the salivary

biomarkers. Significant effects were confirmed using the models adjusted for selected demographic characteristics that had medium size Spearman's correlations ($|\rho|>0.3$) with the biomarker outcome metric.

## Results

All participants (n = 12) completed at-home saliva collection, 4x/day for four consecutive days as instructed (100% adherence). All participants recorded the dates and times for the saliva collection on their collection kits (100% adherence). Each donated saliva sample had sufficient volume for the assays to be performed for each analyte in duplicate. All samples returned assay values within range; the mean values were 0.27 ug/dL (SD 0.26) for cortisol and 148.0 U/mL (SD 114.3) for sAA. The mean EMA response rate was 82.8% (SD 16.3), and the mean number of EMA responses per day was 4.0 (SD 1.2) out of a possible maximum of five per day.

Participant characteristics and descriptive statistics from the surveys, anthropometric and clinical data are presented in Table 1. The mean age was 43.4 years (SD 7.73). The majority of the participants worked full-time. More than half of the participants had an annual income <$60,000. The mean CES-D score was 21.08 (SD 8.36) and about 66% showed a 16 or higher score of CES-D, indicating probable clinical depression. The mean Black racial identity score was 5.21 (SD 1.46), indicating that the majority of the participants self-defined Black race as a core part of their self-concept. About 42% of the participants were obese with a mean BMI of 34.19 kg/cm$^2$.

### Bivariate analyses on surveys, salivary biomarkers and Ecological Momentary Assessment

Table 2 shows the bivariate correlations between the baseline sample characteristics, average levels and scores of salivary cortisol, sAA, and the EMA-reported daily racial discrimination variables; Table 3 shows the differences by sex. A higher educational level was significantly associated with higher sAA levels at wakening ($\rho = 0.69$, $p = 0.01$). Higher systolic ($\rho = 0.77$, $p<0.01$) and diastolic ($\rho = 0.68$, $p = 0.01$) blood pressures were associated with greater AUC of sAA. Lower levels of subjective social status ($\rho = -0.69$, $p = 0.01$) or less alcohol consumption ($\rho = -0.61$, $p = 0.03$) were associated with higher sAA levels at wakening. Lower subjective social status was associated with blunted sAA awakening responses ($\rho = 0.62$, $p = 0.03$). Major lifetime discrimination was associated with daily EMA-reported racial discrimination ($\rho = 0.62$, $p = 0.03$).

### Within-person analyses of daily racial discrimination and salivary stress biomarkers

Intraclass correlation coefficients (ICCs), the proportion of total variance of the salivary stress biomarker outcomes explained by between-person levels, were 0.58 (AUC), 0.29 (awakening response), 0.13 (at wakening), and 0.09 (diurnal slope) for cortisol, and 0.40 (AUC), 0.62 (Awakening Response), 0.66 (at wakening), and 0.50 (diurnal slope) for sAA, respectively.

Tables 4 and 5 show within-person effects of racial discrimination, microaggressions, and negative affects on salivary cortisol and sAA indexes, respectively. The previous day's racial discrimination was significantly associated with the next day cortisol level at wakening ($\beta = 0.81$, SE = 0.21, $p<0.01$) and diurnal slope ($\beta = -0.85$, SE = 0.23, $p<0.01$), indicating the lagged effect of the overt racial discrimination on cortisol. Also, microaggressions were significantly associated with a diurnal cortisol slope in the same day, indicating that on the day when people reported more microaggressions than usual, a flatter diurnal slope of cortisol was observed ($\beta$

**Table 1. Baseline characteristics of participants (N = 12).**

| Category | Mean or N | SD or% |
|---|---|---|
| Age (years) | 43.4 | 7.7 |
| Woman | 8 | 66.6 |
| Working Full-Time | 10 | 83.3 |
| Working Part-Time | 2 | 16.7 |
| Annual income | | |
| $0-$39,999 | 2 | 16.7 |
| $40,000-$59,999 | 6 | 50.0 |
| $60,000-$79,999 | 1 | 8.3 |
| $80,000-$99,999 | 2 | 16.7 |
| >$100,000 | 1 | 8.3 |
| Education | | |
| Some high school | 1 | 8.3 |
| Vocational /technical school | 1 | 8.3 |
| Some college | 5 | 41.7 |
| College graduate | 5 | 41.7 |
| Body mass index (BMI) (kg/m$^2$) | 34.1 | 11.4 |
| 18.5–24.9 (normal) | 3 | |
| 25–29.9 (overweight) | 4 | |
| 30–34.9 (Class 1 obesity) | 0 | |
| 35–39.9 (Class 2 obesity) | 1 | |
| >40 (Class 3 obesity) | 4 | |
| Blood pressure, systolic (mmHg) | 123.0 | 16.1 |
| Blood pressure, diastolic (mmHg) | 82.5 | 13.0 |
| Total body fat (%) | 38.8 | 14.2 |
| Visceral fat (%) | 10.8 | 5.3 |
| Race-related event (0–22) | 11 | 6.3 |
| Major life discrimination (0–9) | 5.1 | 2.5 |
| Racial identity (1–7) | 5.2 | 1.5 |
| Depression (by CES-D) (0–60) | 21.1 | 8.5 |
| Subjective social status (1–10) | 7.1 | 2.4 |
| Smoking | | |
| No | 11 | 91.7 |
| Alcohol consumption | | |
| Never | 3 | 25 |
| Monthly or less | 5 | 41.7 |
| 2–4 times a month | 4 | 33.3 |

Note. Center for Epidemiological Studies Depression Scale (CES-D).

= -0.50, SE = 0.17, $p<0.01$). The within-person analysis showed marginal relationships between more microaggressions and greater AUCg of cortisol ($\beta = 0.40$, SE = 0.21, $p = 0.06$) and higher cortisol levels at wakening ($\beta = 0.40$, SE = 0.20, $p = 0.05$).

Negative affect was positively associated with the diurnal sAA slope in the same day ($\beta = 0.58$, SE = 0.16, $p<0.01$). However, none of the relationships between negative affect and cortisol indexes were significant.

Although some covariates in Table 1, such as age, income, education, depression, blood pressure, subjective social status and alcohol consumption had significant bivariate

**Table 2. Spearman correlations among baseline characteristics, average scores of the EMA survey, and salivary stress biomarkers (n = 12).**

| | EMA Surveys | | Salivary Stress Biomarkers | | | | | | | |
|---|---|---|---|---|---|---|---|---|---|---|
| | Racial discrimination | Microaggression | Cortisol (AUCg) | Cortisol At wakening | Cortisol awakening response | Cortisol diurnal slope | sAA (AUCg) | sAA At wakening | sAA awakening response | sAA diurnal slope |
| Age | -0.47 | -0.34 | -0.35 | -0.38 | -0.06 | 0.43 | 0.28 | -0.41 | 0.43 | 0.02 |
| Body Mass Index | -0.01 | -0.06 | -0.49 | 0.10 | -0.49 | -0.17 | 0.33 | 0.29 | -0.16 | 0.06 |
| Depression | 0.45 | -0.01 | -0.03 | -0.10 | 0.19 | 0.00 | -0.15 | -0.51 | 0.27 | -0.07 |
| Annual Income | -0.18 | -0.44 | 0.19 | 0.24 | 0.34 | -0.38 | -0.12 | 0.48 | **-0.64 | -0.56 |
| Education | -0.15 | 0.11 | 0.11 | 0.38 | 0.03 | -0.42 | -0.06 | **0.69 | -0.41 | -0.41 |
| Systolic Blood Pressure | 0.37 | -0.17 | 0.13 | 0.41 | -0.21 | -0.17 | **0.77 | 0.17 | -0.12 | 0.14 |
| Diastolic Blood Pressure | 0.30 | -0.39 | -0.09 | 0.21 | -0.21 | -0.02 | **0.68 | -0.05 | 0.18 | 0.31 |
| Total Fat | 0.07 | 0.38 | -0.13 | 0.27 | -0.13 | -0.41 | 0.21 | 0.36 | 0.00 | -0.17 |
| Racial Identify | 0.15 | 0.22 | 0.01 | 0.46 | -0.38 | -0.43 | 0.13 | 0.20 | 0.04 | 0.54 |
| Subjective Social Status | -0.12 | -0.38 | -0.40 | -0.28 | -0.16 | 0.30 | 0.25 | **-0.69 | **0.62 | 0.17 |
| Smoking | 0.50 | 0.13 | 0.31 | -0.22 | 0.39 | 0.31 | -0.48 | -0.22 | -0.04 | -0.22 |
| Alcohol Consumption | -0.05 | -0.12 | -0.21 | -0.45 | 0.01 | 0.33 | -0.39 | **-0.61 | 0.49 | 0.21 |
| Past Race-Related Events Scale | 0.51 | 0.52 | 0.26 | 0.31 | -0.01 | -0.43 | 0.00 | -0.09 | 0.17 | 0.32 |
| Major Life Discrimination scale | **0.62 | 0.41 | 0.20 | 0.36 | -0.15 | -0.49 | 0.04 | -0.37 | 0.24 | 0.20 |

Note. Ecological Momentary Assessment (EMA); Depression was measured by the Center for Epidemiological Studies Depression Scale (CES-D); Area Under the Curve ground (AUCg); salivary alpha-amylase (sAA).

** Indicates $p$-value <0.05.

relationships with biometric outcomes, none of the covariates were significant when included in the multilevel models.

The diurnal cortisol rhythm in the participants followed the typical pattern found in adults (Fig 1): high morning wake levels, increasing to a peak 30 minutes after wake and then gradually decreasing across the day. Fig 1A shows the mean cortisol diurnal level, comparing those reporting at least one episode of racial discrimination (solid line) with those reporting none

**Table 3. Baseline characteristics, average scores of the EMA survey, and salivary stress biomarkers by sex (n = 12).**

| | EMA Surveys Mean (SD) | | Salivary Stress Biomarkers Mean (SD) | | | | | | | |
|---|---|---|---|---|---|---|---|---|---|---|
| | Racial discrimination | Microaggression | Cortisol (AUCg) | Cortisol At wakening | Cortisol awakening response | Cortisol diurnal slope | sAA (AUCg) | sAA At wakening | sAA awakening response | sAA diurnal slope |
| Male | 0.50 (0.79) | 25.0 (10.1) | 3.55 (1.98) | 0.36 (0.18) | 0.01 (0.04) | -0.01 (0.01) | 2.33 (10.44) | 108 (87) | -7.3 (33.1) | 2.94 (9.42) |
| Female | 0.88 (1.33) | 50.4 (21.6) | 3.95 (1.56) | 0.41 (0.18) | 0.02 (0.08) | -0.02 (0.01) | 2.30 (10.56) | 170 (130) | -21.8 (35.7) | -0.77 (10.09) |
| Wilcoxon Test ($p$-value) | 0.73 | 0.07 | 0.67 | 0.46 | 0.67 | 0.22 | 0.93 | 0.46 | 0.56 | 0.37 |

Note. Ecological Momentary Assessment (EMA); Area Under the Curve ground (AUCg); salivary alpha-amylase (sAA).

**Table 4. Within-person effect of racial discrimination, microaggressions, and negative affect on cortisol.**

| Predictor | Cortisol | | | |
|---|---|---|---|---|
| | Coefficient ± SE (*p*-value) | | | |
| | AUCg | Awakening response | At wakening | Diurnal slope |
| **Previous Day** (lagged effect) | | | | |
| Racial Discrimination | 0.25±0.21 (.28) | **-0.52±0.24 (.06)** | **0.81±0.21 ($<$ .01)** | **-0.85±0.23 ($<$ .01)** |
| Microaggression | 0.14±0.26 (.60) | 0.32±0.30 (.31) | -0.27±0.25 (.30) | 0.25±0.28 (.39) |
| Negative Affect | -0.01±0.23 (.95) | -0.07±0.26 (.79) | 0.01±0.22 (.96) | -0.01±0.25 (.97) |
| **Same day** | | | | |
| Racial Discrimination | 0.10±0.15 (.51) | 0.17±0.18 (.36) | -0.07±0.18 (.69) | 0.24±0.17 (.17) |
| Microaggression | **0.40±0.21 (.06)** | -0.12±0.22 (.61) | **0.40±0.20 (.05)** | **-0.50±0.17 ($<$ .01)** |
| Negative Affect | 0.06±0.17 (.74) | 0.01±0.20 (.94) | 0.08±0.19 (.67) | -0.15±0.17 (.39) |

Note. Area Under the Curve ground (AUCg).

(dotted line). Fig 1B shows the mean cortisol level comparing those reporting more microaggressions (solid line) with those reporting less microaggressions (dotted line) using a cut-off value of the median. In both Fig 1A and 1B, dotted lines showed lower cortisol levels at wakening and 30-minute post-awakening, compared to the solid lines.

## Discussion

We examined whether the daily racial discrimination/microaggression that we measured in real time using EMA surveys were associated with the ANS and HPA axis activities in healthy Black adults in a natural living environment. Our participants reported overt racial discrimination less frequently (Fig 1. Mean frequency of daily racial discrimination, 0.61 per day) than the microaggressions. We found that microaggressions were significantly associated with a flatter diurnal cortisol slope on the same day. Also, more overt racial discrimination in the previous day was significantly associated with higher cortisol levels at wakening and a flatter diurnal cortisol slope the next day. Our findings were similar to the studies showing the concurrent and lagged effects of racial discrimination on the HPA activity. That is, more episodes of daily racial discrimination were associated with greater total daily cortisol output (AUC) in racially minoritized groups [6, 48]. Also, the lagged effects of racial discrimination

**Table 5. Within-person effect of racial discrimination, microaggressions, and negative affect on salivary alpha-amylase.**

| Predictor | Salivary alpha-amylase | | | |
|---|---|---|---|---|
| | coefficient ± SE (*p*-value) | | | |
| | AUCg | Awakening response | At wakening | Diurnal slope |
| **Previous day** (lagged effect) | | | | |
| Racial Discrimination | 0.00±0.22 (.99) | 0.18±0.11 (.15) | -0.03±0.11 (.79) | -0.22±0.15 (.17) |
| Microaggression | 0.00±0.25 (.99) | -0.37±0.20 (.10) | 0.19±0.20 (.37) | -0.18±0.18 (.37) |
| Negative Affect | 0.03±0.23 (.88) | 0.11±0.13 (.45) | -0.16±0.13 (.25) | **0.58±0.16 ($<$ .01)** |
| **Same day** | | | | |
| Racial Discrimination | -0.16±0.17 (.36) | 0.03±0.15 (.83) | -0.00±0.14 (.97) | 0.01±0.17 (.93) |
| Microaggression | -0.16±0.21 (.46) | -0.05±0.22 (.81) | 0.27±0.22 (.23) | -0.10±0.22 (.65) |
| Negative Affect | -0.12±0.19 (.53) | 0.21±0.17 (.25) | -0.00±0.17 (.97) | 0.13±0.19 (.48) |

Note. Area Under the Curve ground (AUCg).

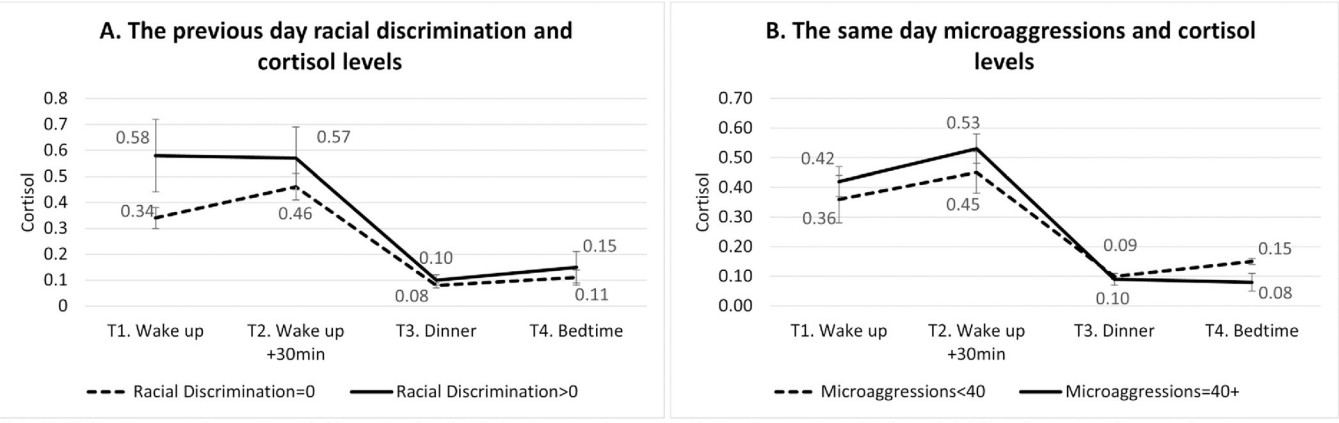

Note: Participants reported on average 0.61 overt racial discrimination experiences per day and most reported substantial daily microaggression experiences.

**Fig 1. Mean cortisol level by the previous day racial discrimination and the same day microaggressions.** (A) The previous day racial discrimination and cortisol levels (B) The same day microaggressions and cortisol levels. Note: Participants reported on average 0.61 overt racial discrimination experiences per day and most reported substantial daily microaggression experiences.

that we found were consistent with the study showing that increased microaggressions predicted greater AUC the subsequent week in Black and Latino young adults [31].

With the expectation that sAA would be linked to cardiovascular measures as a downstream measure of autonomic activity [49], we found significant correlations between sAA levels and systolic and diastolic blood pressure. However, the relationships of sAA levels with racial discrimination and microaggressions were not significant in the multilevel models. This may be explained by our small sample size with predominantly depressed participants. For example, studies have shown that higher levels of negative affective states (e.g., depression, anxiety) are associated with higher baseline sAA, which was in turn related to lesser sAA change after exposure to an acute stressor (i.e., ceiling effect) [49]. Also, although we used EMA surveys for real time measures, temporal proximity in collecting saliva samples and measuring racial discrimination in a natural living environment might have been relatively distant. In a study of Turkish immigrants that examined the effects of acute stress related to discriminations in a lab setting [50] and other intervention studies [51, 52], sAA was a more sensitive measure than cortisol in close temporal proximity—before and after each intervention session. These findings suggest that sAA may be uniquely sensitive to measure the effect of interventions that aim to mitigate stress-related health outcomes and also demonstrate the importance of including sAA measures in assessment of intervention effects focusing on stress responses [19].

Studies have shown that the activities of the HPA axis, which are characterized by elevations in evening cortisol, are significantly associated with poor health outcomes such as incident diabetes [7, 53]. Similarly, unusually high morning cortisol levels have been associated with greater insulin resistance and decreased beta cell function, suggesting that the alterations in cortisol may precede deleterious changes in glucose metabolism [54]. Also, lower morning salivary cortisol, higher bedtime cortisol levels and a flatter diurnal cortisol have been observed in Blacks, independent of socioeconomic status [55, 56]. Studies on sAA are still limited, but a growing number of studies have shown the ANS activity measured by diurnal sAA patterns in individuals experiencing chronic stress or with stress-related disease [18]. For example, diurnal sAA profiles of Bosnian war refugees with posttraumatic stress disorder did not show the expected decrease in post-awakening sAA levels [57]. Therefore, our study findings showing the effect of racial discrimination on ANS and HPA axis activities may have implications for stress related health disparities in Blacks.

The present study has several limitations. It is possible that we may have not captured the peaks in cortisol or sAA levels by collecting saliva only four times per day as peaks in cortisol occur about 15–20 minutes after peaks in perceived stress [58]. Studies using different protocols (e.g., concurrent stress assessment and retrospective measure covering the past hours), however, were equally acceptable [58]. The close timing of saliva collection to its corresponding assessment of racial discrimination using event-contingent sampling (i.e., participants report to EMA whenever a discrimination event occurs) may increase subject burden. Also, the event-contingent sampling may not be able to accurately measure the events if participants forget to report them (i.e., missing EMA). EMA minimizes recall bias and errors, but it is also possible that recording microaggressions may have influenced HPA or ANS activities by heightening vigilance to discrimination from the repetitive assessments involved in EMA.

The small sample size in our study did not allow for sensitivity analysis by sex, which may have influenced ANS and HPA axis activities [19]. Given the small sample size and nature of these data, we cannot determine the extent to which difference might be due to other between-person effects, biological factors (e.g., genetics, comorbidities) versus social stress factors such as racial discrimination. That is, wakeup cortisol levels have been shown to be influenced by genetics and chronic stress, whereas bedtime levels tend to be more reflective of current social stressors [59]. Also, the majority of our participants reported racial discrimination/ microaggression as the main and frequent sources of stress rather than stress from daily hassles (see Supporting information files), which has had less impact on stress responses [58]. Future studies with a large sample size would be helpful to expand our knowledge by teasing out the type of stress and its effect on biological stress responses.

The present study conducting within-person analyses helps control for the influence of stable individual factors and other confounders; however, the study results can also be biased by omitted time-varying factors that may change day-to-day [31]. We did not measure other emotional states such as high arousal positive emotions (e.g., excitement), which may have limited the interpretation of the roles of affective states, particularly on ANS activity [60]. Also, the majority of our participants showed high levels of depressive symptoms, which may have influenced the associations with perceived racial discrimination or negative affect. Future studies need to examine differential associations between cortisol and sAA diurnal profiles and various affective states by age, sex, race/ethnicity, chronic illness comorbidities, and other types of psychosocial stress.

Despite these limitations, the present study provides valuable insights into our understanding of the within-person effects of real-time racial discrimination on concurrent and lagged diurnal cortisol and sAA profiles. We measured the stress response in a comprehensive way, using subjective measures with EMA and ANS, and HPA axis indicators. Using real-time, prospective EMA data allowed for an idiographic approach that examined the effects of racial discrimination and microaggressions in comparison to one's own average level and may have minimized retrospective reporting bias. Lastly, high adherence to the intensive study protocol and multiple days of saliva sample collection in a natural living environment increased the ecological validity in our study.

Racial discrimination and race-related stress are intertwined with social determinants of health rooted in structural racism that result in uneven access to quality school, wealth accumulation, better neighborhoods and quality medical care [1]. The present study examined whether perceived racial discrimination affects diurnal cortisol and sAA rhythms among healthy Black adults. However, future research is also needed that incorporates the multi-level (micro and macro) societal factors of racial health disparities and elucidates the mechanisms of the stress-related health disparities that racially minoritized communities experience.

In conclusion, the results of the current study highlight the potential utility of EMA and salivary cortisol and sAA in studying the effects of daily racial discrimination and microaggressions on biological stress responses, ANS, and HPA axis activities among healthy Black adults. Individuals' racial discrimination or microaggression experiences deeply rooted in structural racism may have concurrent or lagged effects on the HPA axis activities, but its relationship with sAA or negative affect needs to be further examined. Further studies with a large sample are needed to confirm the observed findings in light of our study's limitations and understanding of the physiological, psychological, and social mechanisms of stress reactivity.

## Supporting information

**S1 Data.**
(XLSX)

**S2 Data.**
(XLSX)

**S3 Data.**
(XLSX)

## Author Contributions

**Conceptualization:** Soohyun Nam, LaRon E. Nelson, Douglas A. Granger.

**Data curation:** Soohyun Nam.

**Formal analysis:** Sangchoon Jeon, Garrett Ash.

**Funding acquisition:** Soohyun Nam.

**Investigation:** Soohyun Nam, Douglas A. Granger.

**Methodology:** Sangchoon Jeon, Soo-Jeong Lee, Douglas A. Granger.

**Project administration:** Soohyun Nam.

**Supervision:** Soohyun Nam, Douglas A. Granger.

**Writing – original draft:** Soohyun Nam.

**Writing – review & editing:** Soohyun Nam, Soo-Jeong Lee, Garrett Ash, LaRon E. Nelson, Douglas A. Granger.

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
