## [Decision Letter · Decision Letter 0]

23 May 2022

PONE-D-22-05171Real-time racial discrimination, affective states, salivary cortisol and alpha-amylase in Black adultsPLOS ONE

Dear Dr. Nam,

Thank you for submitting your manuscript to PLOS ONE. After careful consideration, we feel that it has merit but does not fully meet PLOS ONE’s publication criteria as it currently stands. Therefore, we invite you to submit a revised version of the manuscript that addresses the points raised during the review process.

We look forward to receiving your revised manuscript.

Kind regards,

Christopher D. Lynn, Ph.D.

Academic Editor

PLOS ONE

Journal Requirements:

Additional Editor Comments (if provided):

I agree with the suggestions of the reviewers, though I consider all of them relatively minor and straightforward issues to address. Please address all reviewer comments in submitting revised manuscript.

Reviewers' comments:

Reviewer's Responses to Questions

**Comments to the Author**

1. Is the manuscript technically sound, and do the data support the conclusions?

Reviewer #1: Yes

Reviewer #2: Yes

2. Has the statistical analysis been performed appropriately and rigorously? 

Reviewer #1: Yes

Reviewer #2: Yes

3. Have the authors made all data underlying the findings in their manuscript fully available?

Reviewer #1: Yes

Reviewer #2: Yes

4. Is the manuscript presented in an intelligible fashion and written in standard English?

Reviewer #1: Yes

Reviewer #2: Yes

5. Review Comments to the Author

Reviewer #1: The manuscript by Nam and colleagues assesses the effects of racial discrimination on salivary stress responses and affective surveys in real-time. Few studies assess such measures in real-time. Much of the prior work in this area has been conducted retrospectively. The prior day’s self-reported microagressions correlated with cortisol at the time of wakening and the diurnal slope for the day, supporting the notion that concurrent salivary and survey assessment can be used to assess such measures in real-time. The use of the mEMA app is rigorous and several intriguing correlations are revealed. The within-subjects design is a theoretical improvement over past between-subjects designs. Some aspects should be improved prior to publication including the use of the term “dysregulation” which is sometimes used to describe typical HPA axis activation, clarity for the stated hypotheses, some amendments to chosen analyses/figures, and acknowledgement of additional limitations.

ABSTRACT

1. Beta weights and p values are given, however, these are not necessary or particularly descriptive. All p values are assumed to be significant. An r or R^2 would be easier to interpret.

INTRODUCTION

2. It is said that racial discrimination is found to dysregulate the ANS or HPA axis. However, is there evidence for “dysregulation” per se? Dysregulation typically implies a long-term disruption in the capacity for the HPA axis to sufficiently or appropriately activate and restore homeostasis (such as the loss of HPA activity in general adaptation syndrome wherein chronic activation blunts responsivity). It should be clarified that acute insults “activate” the HPA axis and evidence should be provided as to whether a lifetime of racial bias truly “dysregulates” the HPA axis.

3. The reasons for sAA to reflect ANS function should be described (i.e. parotid stimulation by locus coeruleus).

4. It is argued by authors that one of the reasons prior literature has not found a relationship between ANS or HPA axis biomarkers is that studies have been retrospective, thereby accounting for a lifetime of racial bias as opposed to real-time assessment. If it is the case that lifetime racial experiences do not predict ANS or HPA axis function, then this would seem to argue against a true “dysregulation” of the ANS and HPA axis in favor of more immediate changes that may be reversible. Authors should consider this in their interpretation.

5. Why was a blunted wakening cortisol response hypothesized when it has been previously observed that peer rejection increases wakening cortisol?

RESULTS/DISCUSSION

6. Fig. 1 needs to include a measure of variance for each timepoint

7. The correlation with sex is a confusing to interpret. Typically, discrete variables such as sex are not appropriate for continuous inferentials. Sex should really be assessed via t-test. It is suggested to remove the current correlations for this variable in favor of t-tests.

8. “Dysregulation” of the HPA axis is said to be associated with poor health outcomes, but again this wording should be justified.

9. It would be useful to include the “stress from daily hassles” in a supplement.

10. Authors should be careful not to over-interpret cortisol measures. For example, higher salary was associated with

11. Did authors consider that recording microaggressions might heighten awareness to them? Some reactivity to the recording process would be expected, perhaps increasing the HPA axis effect of microaggressions in the current study. This should be considered/discussed.

MINOR COMMENTS

1. Some grammatical errors are noted throughout the manuscript (e.g., inappropriate tense, missing articles such as “the” or “a”, etc). The manuscript should be carefully proofread prior to resubmission.

2. In Table 1, it should read equal to or greater than the top values for salary and BMI.

Reviewer #2: I’ve completed my review PONE-D-22-05171 “Real-time racial discrimination, affective states, salivary cortisol and alpha-amylase in Black adults”. The authors, using longitudinal data from 12 health Black adults, examined the association between racial discrimination and stress response system (cortisol and alpha amylase). The findings revealed that previous day racial discrimination was associated with cortisol; also, microaggressions was associated with cortisol during the same day. Although this study moves research on racial discrimination and objective measures of stress response forward, there is an issue of concern worth noting.

I would like for the authors to connect racism, specifically antiBlack racism, to discrimination. In the current form, discrimination can be reduced to a few “bad apples” instead of a larger system of racism, especially how discrimination and microaggression. For example, if racism conceptualized as both ideology (e.g., attitudes about race and racial inequality) and structure (e.g., laws, policies, etc.), then discrimination can be framed as part of the larger system of racism. This context should be made apparent in both the introduction and the discussion. To help the authors accomplish this goal, I recommend the following papers, especially the Golash-Boza paper:

Doane, A. (2017). Beyond color-blindness:(Re) theorizing racial ideology. Sociological Perspectives, 60(5), 975-991.

Golash-Boza, T. (2016). A critical and comprehensive sociological theory of race and racism. Sociology of race and ethnicity, 2(2), 129-141.

6. PLOS authors have the option to publish the peer review history of their article (what does this mean?). If published, this will include your full peer review and any attached files.

Reviewer #1: No

Reviewer #2: No

---

## [Author Response · Author response to Decision Letter 0]

8 Jul 2022

See the attached file, "Responses to the Reviewers"

---

## [Editor Report · Decision Letter 1]

3 Aug 2022

Real-time racial discrimination, affective states, salivary cortisol and alpha-amylase in Black adults

PONE-D-22-05171R1

Dear Dr. Nam,

We’re pleased to inform you that your manuscript has been judged scientifically suitable for publication and will be formally accepted for publication once it meets all outstanding technical requirements.

Kind regards,

Christopher D. Lynn, Ph.D.

Academic Editor

PLOS ONE

Additional Editor Comments (optional):

Thank you for addressing the requested revisions. I am now recommending your manuscript for publiciation.
---

## [Editor Report · Acceptance letter]

22 Aug 2022

PONE-D-22-05171R1 

Real-time racial discrimination, affective states, salivary cortisol and alpha-amylase in Black adults 

Dear Dr. Nam:

I'm pleased to inform you that your manuscript has been deemed suitable for publication in PLOS ONE. Congratulations! Your manuscript is now with our production department. 

Kind regards, 

on behalf of

Dr. Christopher D. Lynn 

Academic Editor

PLOS ONE